# A simple step heating approach for wall surface temperature estimation in the SOlar and LongWave Environmental Irradiance Geometry (SOLWEIG) model

Nils Wallenberg[1], Björn Holmer[1], Fredrik Lindberg[1], Jessika Lönn[1], Erik Maesel[1], David Rayner[2]

[1]Department of Earth Sciences, University of Gothenburg, Gothenburg, Medicinaregatan 7B, 413 90 Göteborg, Sweden

[2]Swedish National Data Service, University of Gothenburg, Gothenburg, Medicinaregatan 18A, 413 90 Göteborg, Sweden

*Correspondence to*: Nils Wallenberg (nils.wallenberg@gvc.gu.se)

**Abstract.** The urban climate is highly influenced by its building geometry, material characteristics, street orientation and high fraction of impermeable surfaces. All of these influence the microclimate and the resulting outdoor thermal comfort. Mean radiant temperature ($T_{mrt}$) is often used as an estimator for heat exposure as it is one of the most important variables governing outdoor human thermal comfort on clear, calm and warm days. The highest values of $T_{mrt}$ are commonly found in front of sunlit facades where a human is exposed to high levels of direct and reflected shortwave radiation from the sun, as well as high levels of longwave radiation emitted from surrounding sunlit walls. As a consequence, outdoor thermal comfort modelling requires accurate simulation of wall surface temperatures ($T_s$).

The aim of this study is to present a step heating approach for calculating wall $T_s$ in the SOlar and LongWave Environmental Irradiance Geometry model (SOLWEIG) and quantify how it influences $T_{mrt}$. This method requires information on material characteristics, i.e. specific heat capacity, density, thermal conductivity, albedo and thickness of the outer layer of the wall, as well as radiation balance at the wall surface, and ambient air temperature. Simulated $T_s$ is compared to observed $T_s$ of two white walls (albedo = 0.5) in Gothenburg, Sweden; one wooden wall and one plaster brick wall. The simulations show high agreement with the 15,394 observations, with $R^2 = 0.93$ and RMSE = 2.09 °C for the wooden wall and $R^2 = 0.94$ and RMSE = 1.94 °C for the plaster brick wall. For the walls presented here, this new parameterization scheme results in differences in $T_{mrt}$ of up to 2.5 °C compared to the previous version of SOLWEIG.

With this new approach, SOLWEIG can be used to evaluate the effect of building materials on outdoor thermal comfort. The speed and accuracy of this approach suggests that it also could be applied in other areas where $T_s$ of walls are important, for example building energy models and urban energy balance models.

## 1 Introduction

The climate in urban areas is highly influenced by its building geometry (Muniz-Gäal et al., 2020; Nasrollahi et al., 2021; Xi et al., 2021), the orientation of streets (Ali-Toudert & Mayer, 2005), material characteristics (Santamouris and Yun, 2020; Adilkhanova et al., 2023; Battista et al., 2023), vegetation (Schibuola, L. & Tambani, C, 2022; Stache et al., 2022) and the relatively high fraction of impermeable surfaces (Arnfield, 2003). Outdoor human thermal comfort is a consequence of the resulting microclimate. On clear, calm and warm days,

the mean radiant temperature ($T_{mrt}$) is one of the most important factors in describing the outdoor thermal comfort of a human, and is used in many thermal comfort indices, e.g. the Physiological Equivalent Temperature (PET) (Mayer & Höppe, 1987; Höppe et al., 1999) and the Universal Thermal Climate Index (UTCI) (Blazejczyk et al., 2010). For UTCI, however, there are shortcomings when using it within the urban canopy layer because of its requirement of wind speed at 10 meters above ground (Lee et al., 2025). $T_{mrt}$ describes the radiant load that a human is exposed to, i.e. the exchange of short- and longwave radiation between the person and its surrounding radiative environment (Höppe, 1992; Thorsson et al., 2007). While spatial differences in air temperature ($T_{air}$) are usually very small in urban environments, the spatial differences in $T_{mrt}$ between sunlit and shaded spaces can be up to 30 °C (Lau et al., 2015). For these reasons, many authors stress the importance of accurate modeling of $T_{mrt}$ (Di Napoli et al., 2020; Gál and Kántor, 2020).

The shortwave radiation that a human is exposed to originates from the sun, with direct, diffuse and reflected components, whereas the longwave radiation originates from the sky, ground, vegetation and building surfaces, e.g. walls. There are numerous models to calculate $T_{mrt}$, including ENVI-met (Bruse and Fleer, 1998), RayMan (Matzarakis et al. 2007; Matzarakis et al. 2010), PALM-4U (Maronga et al., 2020) and The SOlar and LongWave Environmental Irradiance Geometry model (SOLWEIG) (Lindberg et al., 2008), all of which to some extent handle the above mentioned fluxes. Highest values of $T_{mrt}$ are commonly found in front of sunlit walls (Lindberg et al., 2014) where a person would be exposed to high levels of direct and diffuse shortwave radiation as well as comparatively high amounts of reflected shortwave radiation off the wall (Wallenberg et al., 2020). Another highly influential aspect is if the wall is (or has recently been) sunlit, which increases its surface temperature ($T_s$) so that a person located close to the wall is exposed to higher emittance of longwave radiation.

There are several ways of estimating the $T_s$ of a material. A common approach is with Fourier's law of diffusion (e.g. Simon, 2016; Resler et al., 2017), where the $T_s$ of a material is calculated based on its energy balance that is estimated from heat conduction to or from the interior of the material, convection and radiative exposure. Here, thermal properties such as thermal conductivity, specific heat capacity and emissivity of the material are important, information that is not always easily accessible, especially in the interior of the walls. While this approach is accurate and dynamic, it is relatively time-consuming. Another faster approach is the force-restore method (Johnson et al., 1991). The force-restore approach uses two layers, one thin layer at the surface and one deeper layer, where the surface layer is exposed to gain or loss of energy and a deeper layer that restores or dampens the heating at the surface. Other approaches include empirical relationships, where $T_s$ is observed and related to environmental variables, e.g. $T_{air}$ or solar radiation, as in Bogren et al. (2000) and Lindberg et al. (2016). The empirical approach is only accurate for the specific material that it is calibrated on and requires logistically demanding field observations to incorporate new materials in modeling. Potentially, it is also only accurate for the time of measurements, e.g. season, used in the calibration. An additional approach that can be used to estimate the $T_s$ of a material is step heating. Boue & Fournier (2009) used a step heating equation derived from the Dirac heat pulse to estimate $T_s$ of materials. This is a straightforward approach, where the radiation balance at the surface of the material is used together with its thermal effusivity. Here, thermal effusivity (or thermal admittance (see Oke et al., 2017)) describes the ability of a material to absorb or return the thermal energy at its surface, for example between a wall surface and the ambient air.

As mentioned before, a common approach to estimate $T_s$ is to use the concept of Fourier's law of diffusion and divide the wall into several layers and calculate its energy balance. The ENVI-met model uses three layers and calculates the temperature in seven nodes (Simon, 2016), whereas PALM-4U uses four layers and six nodes (Resler et al., 2017; 2021). Both of these models can simulate $T_s$ with high accuracy. The TARGET model uses the force-restore method to estimate wall surface temperatures (Broadbent et al., 2019). Up to now, the SOLWEIG model uses the empirical approach described in Bogren et al. (2000) and Lindberg et al (2016). The simulated $T_s$ of ENVI-met (Simon, 2016) and PALM-4U (Resler et al., 2017; 2021) show high agreement with observations. The $T_s$ of walls simulated with SOLWEIG has not been evaluated specifically. Gal & Kantor (2020) found that $T_{mrt}$ simulated with SOLWEIG was underestimated in sunlit areas and overestimated in shaded areas, and that these offsets could be related to its wall surface temperature parameterization, i.e. the simple empirical approach (Bogren et al., 2000; Lindberg et al., 2016). Wallenberg et al. (2023) also argued that the parameterization of wall surface temperatures in SOLWEIG could result in deviations in simulated emitted longwave radiation from building surfaces. In addition, SOLWEIG not being a full energy balance model, using simplified convection and conduction parameterizations, if included at all, should not be ignored.

The aim of this study is to present and evaluate a new step heating model approach for calculations of $T_s$ of walls in the SOLWEIG model, and how this affects $T_{mrt}$.

## 2 Methods

### 2.1 Surface temperature parameterization

The basis of the $T_s$ simulation is the step heating equation derived from the Dirac heat pulse as described in e.g. Boue & Fournier (2009):

$$T_s = \frac{2\omega}{e}\sqrt{\frac{t}{\pi}} + T_{air} \tag{1}$$

where:

- $\omega$ is the heat flux density (W/m$^2$), i.e. the sum of incoming and outgoing short- and longwave radiation of the wall;
- $e$ is the thermal effusivity (W s$^{0.5}$ m$^{-2}$ K$^{-1}$) of the wall surface material, derived from:

$$e = \sqrt{\lambda \rho C} \tag{2}$$

with $\lambda$ being the thermal conductivity of the material, $\rho$ is the density and $C$ is the specific heat capacity.

- $t$ is the characteristic time in seconds, as described below.
- $T_{air}$ is the air temperature.

The wall surface is considered a semi-infinite solid that has a thickness $L$ and extends infinitely in other directions. The time $t$ in eq. 1 is estimated from the characteristic time that describes the time it takes for the energy to propagate from the outdoor surface of the material to its approximately interior end, i.e. through the thickness $L$.

The characteristic time is calculated according to eq. 3 (Parker et al., 1961; Cape & Lehman, 1963; Xue et al., 1993; Philipp et al., 2019):

$$t = \frac{L^2}{(\pi^2 \kappa)} \tag{3}$$

where $\kappa$ is the thermal diffusivity calculated from $\lambda / \rho C$. With eq. 3 specific heat capacity and density are eliminated, meaning that they have no effect on the estimated $T_s$. That is, if you increase e.g. specific heat capacity you would increase $e$ (eq. 2), but simultaneously increase $t$ (eq. 3) to the extent that $T_s$ is left unchanged.

Since $\omega$ in eq. 1 is dependent on $T_s$ of the previous timestep (outgoing longwave radiation), the initial $T_s$ at timestep 0 is set to $T_{air}$ and the simulation should therefore ideally start a few hours before sunrise when $T_s \approx T_{air}$. After this, $T_s$ at timestep $i$ is set to $T_{s,i-1}$ and $T_{air}$ is $T_{air,i}$. In addition to this, as $\omega$ includes $T_s$ of walls from the previous timestep, eq. 1 is executed twice: first with $T_{s,i-1}$ and then a second time with an updated $T_s$ (outgoing longwave radiation). The following section, 2.2, gives a more detailed description of how short- and longwave radiation is estimated.

**2.2 Calculation of received short- and longwave radiation**

The omega term ($\omega$ in eq. 1) is the sum of incoming and outgoing short- and longwave radiation for the wall surface of interest. It is calculated here with the SOLWEIG model (Lindberg et al., 2008) that is described in more detail in section 2.4.

**2.2.1 Shortwave radiation**

The absorbed shortwave radiation is the sum of the direct ($K_{dir}$), sky diffuse ($K_{diff}$) and reflected ($K_{ref}$) components. These components are calculated as follows:

$$K_{dir} = (1 - \alpha_{wall}) \times I \times Sh \times \zeta \tag{4}$$

$$K_{diff} = (1 - \alpha_{wall}) \times D \times \psi \tag{5}$$

$$K_{ref} = (1 - \alpha_{wall}) \times \left( G \times \alpha_{wall} \times F_b + G \times \alpha_g \times F_g \right) \tag{6}$$

where $I$ is the direct shortwave radiation from the sun, $D$ is the sky diffuse shortwave radiation and $G$ is horizontal global radiation. $\alpha_{wall}$ and $a_g$ are the albedo of the wall and ground surface respectively, $Sh$ is a Boolean value indicating if the wall surface is sunlit or not, $\zeta$ is the angle of incidence (defined below) of $I$ and $\psi$ is the sky view factor at the wall surface. $K_{ref}$ consists of two terms, where the first one refers to shortwave radiation reflected from surrounding building surfaces ($F_b$, defined below) and the second refers to shortwave radiation reflected off the ground ($F_g$, defined below).

The angle of incidence, $\zeta$, is calculated according to eq. 7:

$$\zeta = \cos \eta \cos \theta \cos \varphi + \sin \eta \sin \theta \sin \varphi \tag{7}$$

where $\eta$ is the solar altitude, $\theta$ is the solar azimuth and $\varphi$ is the wall aspect (0° = north facing wall).

### 2.2.2 Longwave radiation

The longwave radiation is the sum of longwave radiation from surrounding sunlit walls ($L_{wall,sun}$), surrounding shaded walls ($L_{wall,sh}$), the sky ($L_{sky}$), reflected ($L_{ref}$) and ground ($L_{ground}$) and are calculated according to equations:

$$L_{wall,sun} = \sigma \varepsilon_w T_{s,sun}^4 F_b (1 - f_{sh}) f_{sun} \tag{8}$$

$$L_{wall,sh} = \sigma \varepsilon_w T_{s,sh}^4 F_b (1 - f_{sh})(1 - f_{sun}) + \sigma \varepsilon_w T_{s,sh}^4 F_b f_{sh} \tag{9}$$

$$L_{sky} = \sigma \varepsilon_{sky} T_{air}^4 \psi \tag{10}$$

$$L_{ref} = (1 - \varepsilon_w)(L_{down} + L_{up}) F_b \tag{11}$$

$$L_{ground} = L_{up} F_g \tag{12}$$

where σ is Stefan-Boltzmann constant, $\varepsilon_w$ and $\varepsilon_{sky}$ (calculated here according to Prata 1996) are the emissivity of the wall and sky respectively, $T_{s,sun}$ and $T_{s,sh}$ are the mean surface temperature of the model domain sunlit and shaded wall surfaces respectively, $F_b$ is the fraction of non-sky surfaces, $f_{sh}$ is the sunlit fraction calculated as from a cylindric wedge (Lindberg et al., 2008) and $f_{sun}$ is the fraction of sunlit non-sky surfaces. $L_{up}$ is calculated according to equation 17 in Lindberg et al (2016):

$$L_{up} = \psi \left( \varepsilon_g \sigma \left( T_{air} + \left( S_b - (1 - S_v)(1 - \tau) \right) \left( T_{s.g} - T_{air} \right) \right)^4 \right) \tag{13}$$

where $\psi$ is sky view factor (SVF), $\varepsilon_g$ is emissivity of the ground, $S_b$ and $S_v$ indicates if the surface is shaded from buildings or vegetation, respectively, $\tau$ is transmissivity of shortwave radiation through the vegetation and $T_{s.g}$ is the surface temperature of the ground.

$L_{down}$ is estimated according to equation 15 in Wallenberg et al. (2023a):

$$L_{down} = \sum_{i=1}^{153} \varepsilon_{p.i} \sigma T_{p.i}^4 \, \varphi_i \cos \xi_i \frac{1}{\pi} \tag{14}$$

where $\varepsilon_{p.i}$ is the emissivity of one of the 153 patches, $T_{p.i}^4$ the temperature of the patch (sky, wall or vegetation), $\varphi_i$ the solid angle of the patch and $\cos \xi_i$ is the angle of incidence.

The fraction of sunlit non-sky surfaces, $f_{sun}$, is calculated based on the wall aspect and solar azimuth and is scaled between 0.2 and 0.8:

$$f_{sun} = 0.2 + \frac{|\varphi - \theta|}{180°} \times 0.6 \tag{15}$$

The scaling is performed to avoid totally sunlit or totally shaded areas. It is reasonable to assume that a built-up area or a street canyon facing the sun is not totally sunlit but would consist of partly shaded areas, depending on its geometry. This relationship is a simplification of reality to maintain computational speed.

An outgoing longwave component is also calculated, first based on the wall $T_s$ of the previous timestep (i-1) and then in a second iteration with its updated $T_s$ (as described in the last paragraph in 2.1):

$$L_{out} = \varepsilon_w \sigma T_s^4 \tag{16}$$

### 2.3 View factors at wall surfaces

Individual vertical pixels are described as voxels, with a resolution equal to the input raster's (see section 2.4). To estimate the amount of short- and longwave radiation that reaches each voxel, view factors are required. SVF ($\psi$) used in equations 5 and 10 is calculated at each voxel height and can never exceed 0.5 as only half of the upper hemisphere is seen. The SVF scheme in SOLWEIG can differentiate between buildings and vegetation, meaning that for each horizontal pixel the amount of sky blocked by buildings and vegetation (canopy), respectively, is estimated. In addition, the SVF scheme calculates if a canopy is in front of a building. Thus, the building view factor ($F_b$), i.e. buildings seen by a voxel is:

$$F_b = 1 - \psi_b - 0.5 \tag{17}$$

where $\psi_b$ is SVF based on buildings at ground level. At ground level it can be assumed that 0.5 of what is seen by a voxel is ground surface. For example, if $\psi_b$ is 0.25 and ground seen is 0.5, the remaining view factor $= F_b = 0.25$, is non-sky. Vegetation view factor, i.e. amount of canopy seen by a voxel, is calculated in a similar way:

$$F_v = 1 - \psi_v - 0.5 \tag{18}$$

where $\psi_v$ is SVF based on vegetation at ground level.

If $F_b$ and $F_v$ are assumed to not change with height, the ground view factor ($F_g$) seen by voxels at higher levels is then:

$$F_g = 1 - F_v - F_b - \psi \tag{19}$$

where $F_g$ is expected to change with a change in $\psi$, i.e. SVF (buildings and vegetation combined) at each voxel.

The calculations of view factors, as with $f_{sun}$ (eq. 13), are simplifications of reality, to reduce computing time. There are shortcomings, for example if a wall surface extends higher than an opposite wall surface. In this case $F_b$ would eventually deviate, when it should rather gradually turn into $F_g$. Similar consequences can be drawn for $F_v$. It is, however, expected that the implications for $T_{mrt}$ at pedestrian level will be small.

### 2.4 Spatial calculation of voxel locations

The locality of each voxel is determined by the shadow casting algorithm in SOLWEIG (Ratti & Richens, 1999). To determine which voxel is visible from which ground pixel, i.e. by a human, the patch methodology presented in Wallenberg et al. (2020; 2023) is utilized.

### 2.5 SOLWEIG simulation setup

SOLWEIG is a 2.5D radiation model commonly used in studies on radiant load of humans (e.g. Thom et al., 2016; Bäcklin et al., 2021; Wallenberg et al., 2023b). The model is available via the Universal Multi-scale Environmental Predictor (UMEP) (Lindberg et al., 2018). Radiant conditions are estimated in SOLWEIG from pixel-based information on ground and building elevation (Digital Surface Model (DSM)) together with meteorological data (at least global shortwave radiation, $T_{air}$ and relative humidity). It is also possible to include pixel-based information on vegetation height (Canopy Digital Surface Model (CDSM)) as well as ground cover. The shadow casting algorithm in SOLWEIG (Ratti and Richens, 2004; Lindberg and Grimmond, 2010; 2011)

utilizes the information on elevation of ground, buildings and vegetation to calculate shadows, including shadows on walls, thus differentiating between shaded and sunlit pixels. SOLWEIG has been evaluated in several studies (e.g. Lindberg et al., 2008; Lindberg and Grimmond, 2011; Kantor et al., 2018; Gal and Kantor, 2020).

Here, we use a DSM and a CDSM at 0.5 m spatial resolution (81x80 pixels), with walls divided into 0.5 m voxels, together with direct and diffuse shortwave radiation, $T_{air}$, relative humidity and mean sea level pressure at 10-minute temporal resolution to calculate the net radiation balance $\omega$ of the wall surfaces. The walls have been divided into 7182 unique voxels. $T_{air}$ and relative humidity was observed with a TinyTag Plus 2 (Tinytag, 2019) located on the northern side of the building (see Section 2.5). The remaining meteorological variables were

retrieved from a weather station on the rooftop of the Department of Earth Sciences, University of Gothenburg (57.6883°, 11.9663°).

**2.6 Field observations and wall descriptions**

Observations were conducted on a three-story building in Gothenburg, Sweden. The wall is built of two different materials, where the ground floor consists of plastered brick ($L = 0.1$ m) while the upper two are made of wood

($L = 0.03$ m) (Fig. 1). Both parts of the wall are painted white. They have the same aspect (154°) but differ due to the different height and SVF and thus, differ in incoming short- and longwave radiation. Wall surface temperature was observed with two Apogee SI-111 Research-Grade Standard Field of View Infrared Radiometer Sensors (Apogee Instruments, 2024a; 2024b) positioned at a slight angle facing upwards and approximately 10 cm from the wall surfaces (see figure 1c-d). The short distance of the sensors to the wall was to avoid micro-shadows and

the angle to avoid self-shadowing. All wall characteristics except for the albedo and emissivity are set according to look-up tables (CIBSE, 2015) and are presented in Table 1. Continuous observations between 2023-05-15 and 2023-08-31 were recorded, thus covering an entire summer with varying weather from entirely clear days (e.g. 12[th] of June) to fully overcast (e.g. 8[th] of August). Ambient $T_{air}$ ranged from 5.7 °C (17th of May 8:20) to 28.4 °C (17[th] of June 11:30) (Tiny Tag Plus 2) and daily average global shortwave radiation ranged from 24 Wm[-2] to 375

225 Wm[-2] (department weather station). Albedo of the walls was assumed to be 0.5 (see e.g. Celniker et al., 2021 for albedos of similar walls). The wall emissivity was set to 0.95 to match the settings of the Apogee SI-111 sensor.

Table 1: Thermal conductivity ($\lambda$), density ($\rho$), specific heat capacity ($C$), thermal diffusivity ($\kappa$) and thermal effusivity ($e$) of the materials studied. Values for $\kappa$ and $e$ are calculated based on $\lambda$, $\rho$ and $C$ retrieved from CIBSE (2015) (tables [1]3.37, [2]3.38 and [3]3.46).

| Wall type | $\lambda$ (W m[-1] K[-1]) | $\rho$ (kg -m[-3]) | $C$ (J kg[-1] K[-1]) | $\kappa$ (m[2] s[-1] 10[-6]) | $e$ (J m[-2] s[-0.5] K[-1]) |
|---|---|---|---|---|---|
| Brick, outer leaf[1] | 0.84 | 1700 | 800 | 0.62 | 1068 |
| Dense plaster[3] (on brick) | 0.57 | 1300 | 1000 | 0.44 | 860 |
| Hardwood (unspecified), dry[2] | 0.17 | 700 | 1880 | 0.13 | 472 |

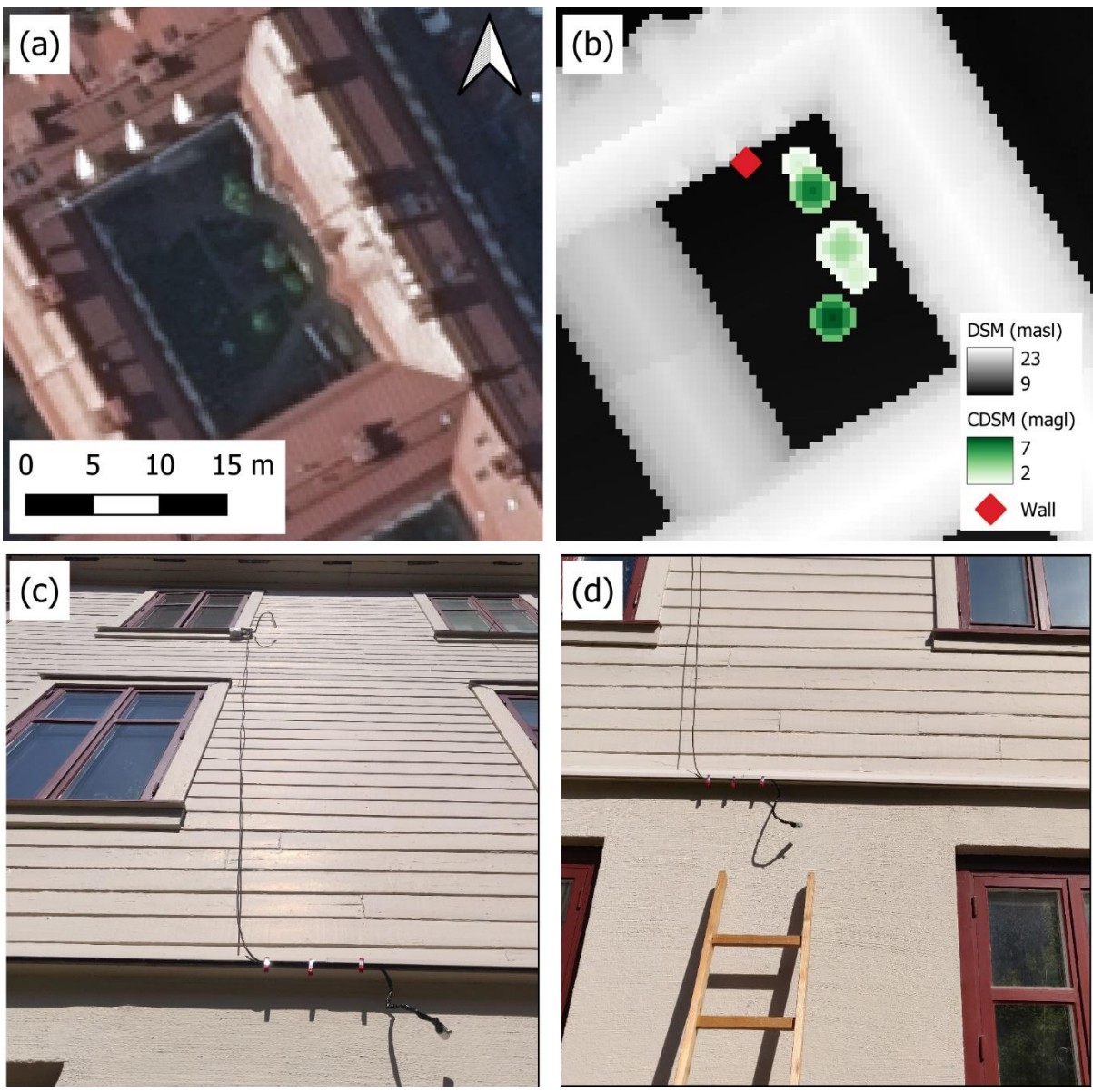

**Figure 1: Orthophoto (© Lantmäteriet) (a) and digital surface model (DSM) and canopy digital surface model (CDSM) (b) of the courtyard surrounding the observed wall. The location of the wall is indicated by a red diamond. The observed walls with sensors can be seen in (c) and (d).**

## 3 Results

Five days of simulated and observed wall $T_s$ for the wooden and plaster brick walls are presented in figure 2a and 2b, respectively. These days are characterized by relatively high $T_{air}$, peaking at between 22-26 °C (figure 2c), and mainly clear weather conditions, with global shortwave radiation ($K_\downarrow$) at solar noon peaking at 860-914 Wm$^{-2}$ (figure 2c). Both walls show good agreement, with the simulated wall $T_s$ consistently following the observed values. However, some deviations are evident, particularly around the time the wall becomes sunlit and in the nighttime. Around sunrise the simulated $T_s$ for the wooden wall are underestimated at 2-6 °C (figure 2a). In the afternoon the simulated cooling rate of the wall is too slow, leading to overestimations of around 1-2 °C that continue into the nighttime. For the last two days, midday temperatures were underestimated by around 5 °C.

The simulated $T_s$ for the plaster brick wall (figure 2b) overestimate up to 2 °C at midday during the first three

245  days. Contrary to the wooden wall, the plaster brick wall underestimates from the evening into nighttime, with 1-3 °C, as an effect of a too rapid simulated cooling rate.

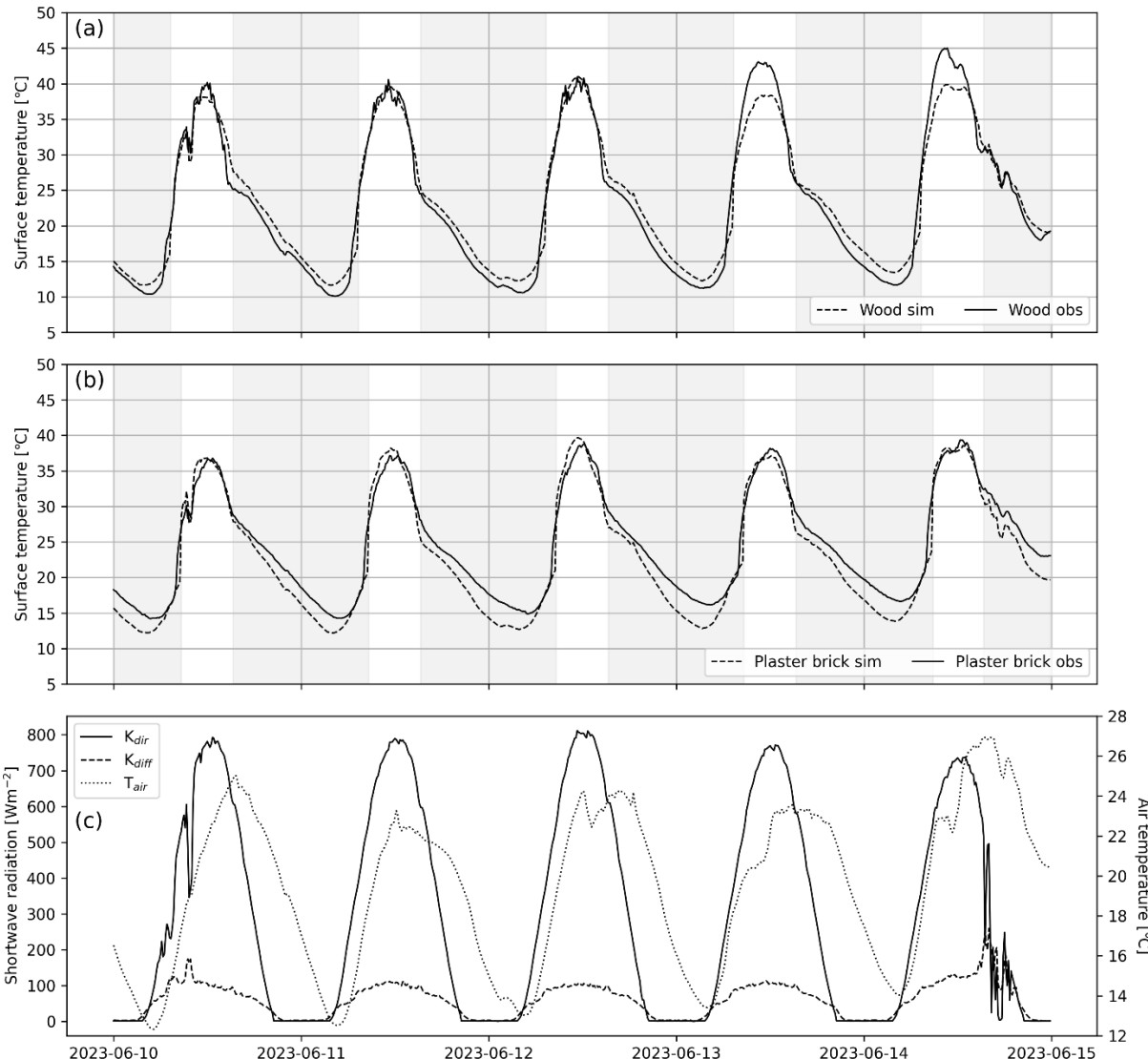

**Figure 2: Plots showing five days of observed (solid) and simulated (dashed) wall surface temperature in 10-minute temporal resolution for (a) the wooden wall and (b) the plaster brick wall. Grey areas show when the wall voxel is in**

**simulated shade. Meteorological forcing data is given in (c), with direct solar radiation ($K_{dir}$) in solid, diffuse solar radiation ($K_{diff}$) in dashed and air temperature ($T_{air}$) in dotted lines.**

Fig. 3 shows the diurnal development for the whole study period (2023-05-15 – 2023-08-31). The top panel (figure 3a) shows $T_s$ for the wooden wall. The mean values of the simulation (dashed blue line) are aligned with the mean of the observations (solid line), with a small underestimation of 1-2 °C between 08:00-13:00. Before and after

these hours, simulations are close to observations with a very small overestimation. The spread of the data can be explained by different weather conditions, ranging from fully overcast, shown as low daytime values, to high values correlating to clear weather conditions with high amounts of incoming solar radiation. Here, it is also noticeable that the simulations (in blue) diverge from the observations, with underestimations during intense solar radiation and overestimations in nighttime.

The plaster brick wall, presented in figure 3b, shows opposite patterns compared to the wooden wall. Here, simulations overestimate with 1 °C in daytime, and underestimates by 2-3 °C in nighttime. These patterns are also visible in the spread of the data, where simulations overestimate during intense solar radiation. There is a narrower spread for the plaster brick wall, compared to the wooden wall, which can be explained by their material characteristics, i.e. the wooden wall reacts much faster to changes in solar radiation.

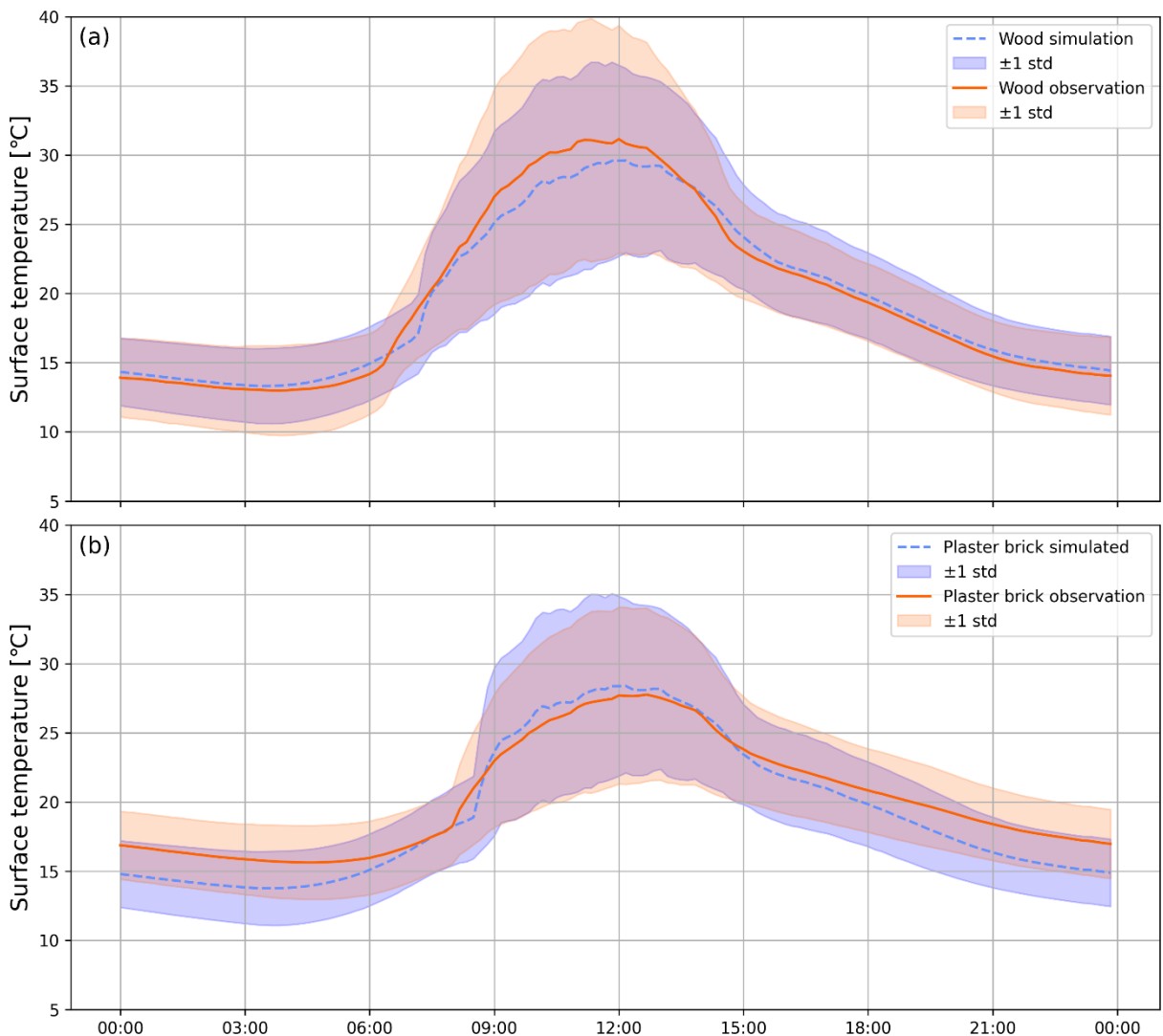

**Figure 3: Plots showing wall surface temperatures from 2023-05-15 through to 2023-08-31 (107 days) for (a) the wooden wall and (b) the plaster brick wall. The dashed and solid lines show mean simulated and observed wall surface temperatures, respectively.**

Scatter plots of 10-minute simulated and observed $T_s$ of the wooden wall, classified into five classes of mean $K_\downarrow$ between 11:00-14:00 are presented in figures 4a-e, with all values given in figure 4f. For example, an entire day will be classified into $K_\downarrow < 150$ Wm$^{-2}$ if the mean $K_\downarrow$ between 11:00 – 14:00 on this day is below 150 Wm$^{-2}$. The five classes are $K_\downarrow < 150$ Wm$^{-2}$ (9 days), 150-300 Wm$^{-2}$ (14 days), 300-450 Wm$^{-2}$, (14 days), 450-600 Wm$^{-2}$ (19 days) and $K_\downarrow > 600$ Wm$^{-2}$ (51 days).

The first class ($K_\downarrow < 150$ Wm$^{-2}$) (figure 4a) shows high correlation between simulations and observations ($R^2 = 0.93$) and low deviation of simulations (RMSE 0.82 °C). The three following classes (figures 4b-d) have lower correlations, partly because days in these classes are subject to partly cloudy weather, compared to the first class

that is subject to overcast conditions. The last class (figure 4e) shows higher agreement between simulations and observations ($R^2$ = 0.95). Overall, with 15,394 observation points (figure 4f), the simulations for the wooden wall correlate well with observed values ($R^2$ = 0.93) and show relatively low RMSE (2.09 °C), indicating that the

model performs well.

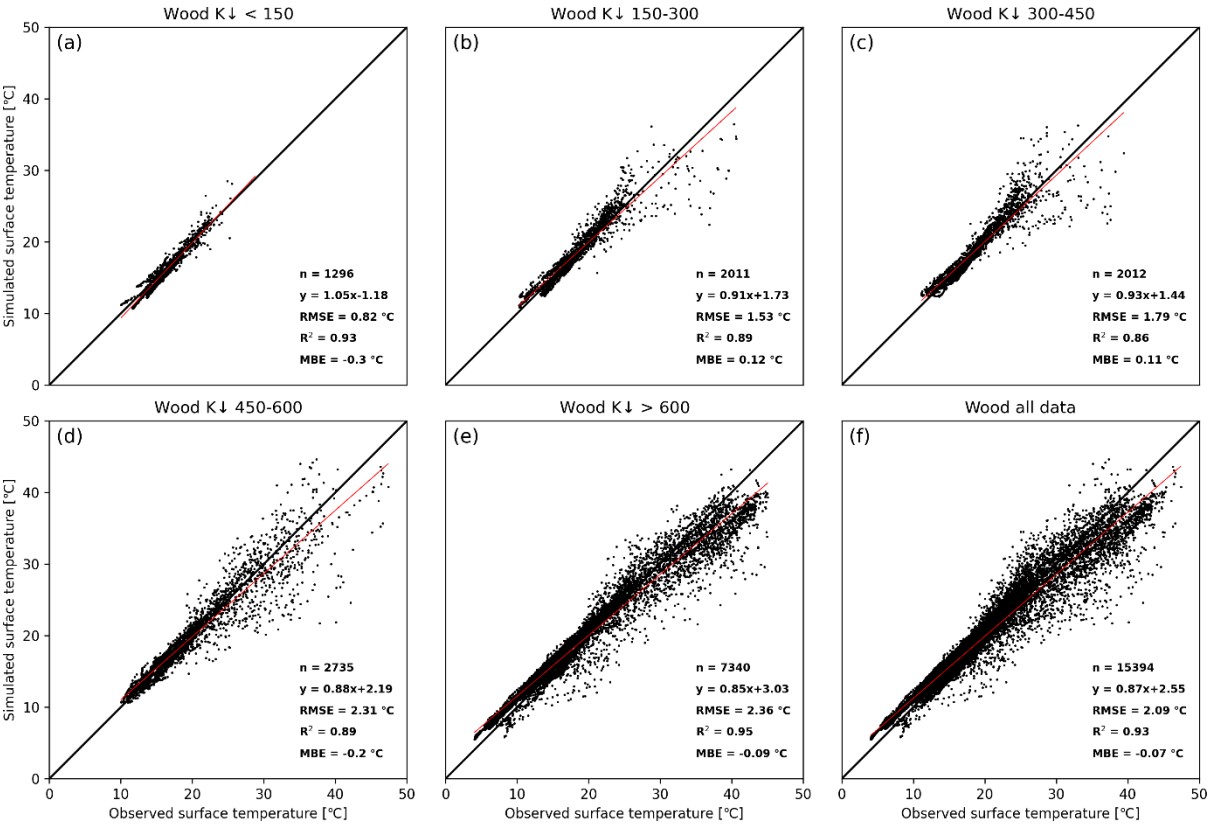

**Figure 4: Scatter plots of 10-minute simulated and observed $T_s$ of the wooden wall, classified according to average incoming global shortwave radiation between 11:00 – 14:00 of (a) $K_↓$ < 150 Wm$^{-2}$, (b) 150-300 Wm$^{-2}$, (c) 300-450 Wm$^{-2}$, (d) 450-600 Wm$^{-2}$, (e) > 600 Wm$^{-2}$, and (f) all data.**

Similar scatter plots to those in figure 4, but for the plaster brick wall, are presented in figure 5. Correlations are generally good for all five classes. Best agreement can be seen in $K_↓$ > 600 Wm$^{-2}$ ($R^2$ = 0.95) (figure 5e), whereas lowest is evident in $K_↓$ < 150 Wm$^{-2}$ (figure 5a) and $K_↓$ 300-450 Wm$^{-2}$ (figure 5c) classes with $R^2$ = 0.86. Highest deviation is found in $K_↓$ 450-600 Wm$^{-2}$ (RMSE = 2.09 °C) (figure 5d), whereas lowest deviation is evident for $K_↓$ < 150 Wm$^{-2}$ (RMSE = 1.46 °C) (figure 5a), which can be explained by low incoming shortwave radiation during

overcast weather conditions, resulting in a $T_s$ of the wall that is possibly close to $T_{air}$. All observations combined (figure 5f), simulated $T_s$ show good agreement with observations ($R^2$ = 0.94) and small deviations (RMSE = 1.94 °C).

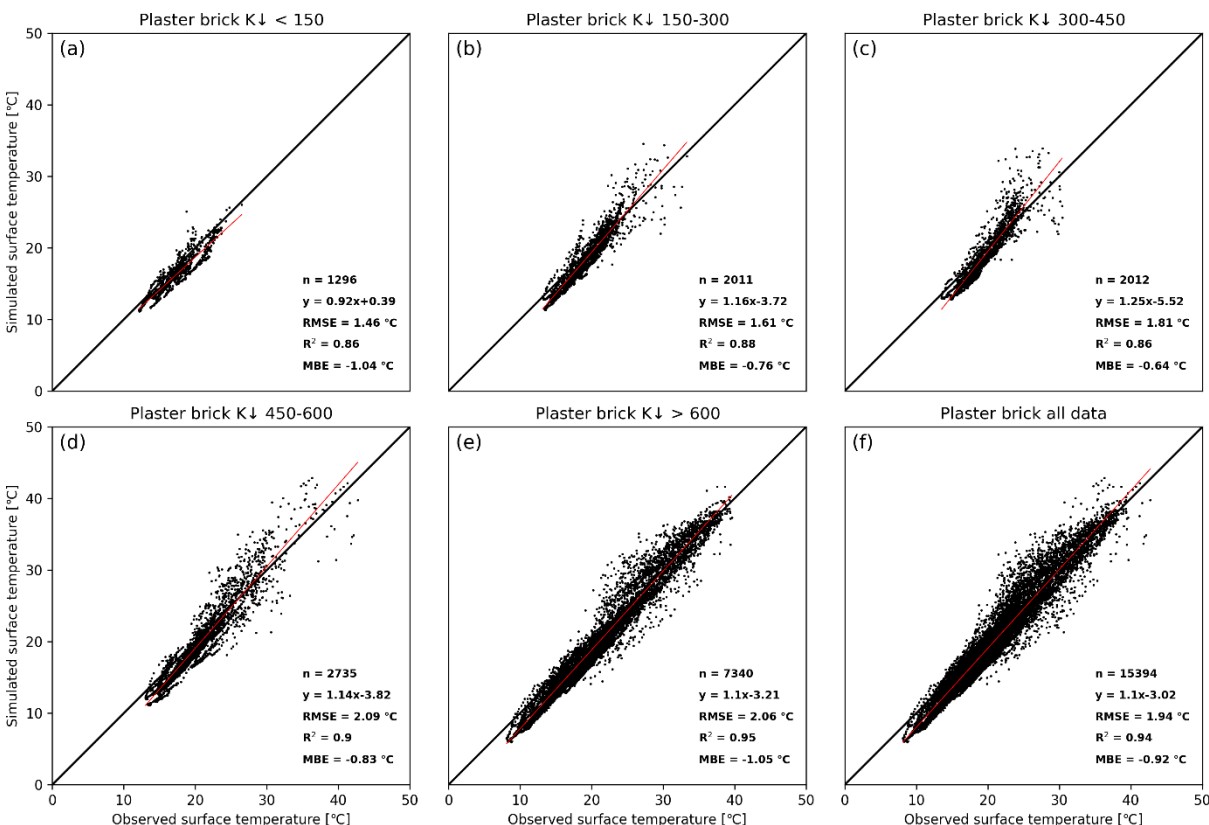

**Figure 5: Scatter plots of 10-minute simulated and observed $T_s$ for the plaster brick wall, classified as for Figure 4 into (a) $K_\downarrow$ < 150 Wm$^{-2}$, (b) 150-300 Wm$^{-2}$, (c) 300-450 Wm$^{-2}$, (d) 450-600 Wm$^{-2}$, (e) > 600 Wm$^{-2}$, and (f) all data.**

Comparisons in $T_{mrt}$ at different timesteps are presented in figure 6 to assess the effect of the new parameterization scheme for wall $T_s$ with the previous method (Lindberg et al., 2008). The simulations are for 2023-06-12 (as shown in Figure 2) and the different timesteps compared are (a) 00:00, (b) 03:00, (c) 06:00, (d) 09:00, (e) 12:00, (f) 15:00, (g) 18:00 and (h) 21:00.

The effect of the new scheme on nighttime $T_{mrt}$ in the courtyard is small but noticeable, around 0.3 °C lower compared to the old version (v2022a) of SOLWEIG (figures 6a-b). At 06:00 in the morning (Figure 6c), when the sun is above the horizon but still at a low elevation, more differences in $T_{mrt}$ appear, with some areas having slightly higher values (0.4 °C), while others have slightly lower (–0.2 °C). At 09:00 (figure 6d), more pronounced differences appear, with areas close to the northwestern corner of the courtyard being up to 2.5 °C warmer compared to SOLWEIG v2022a. Here, the sun reaches most of the areas in this corner as the sun is in the east at around 35° elevation, meaning that the direct shortwave component is close to perpendicular to the walls, resulting in high amounts of radiation reaching these surfaces. This becomes evident when looking at the differences at 12:00 (figure 6e), when incoming shortwave radiation is higher but differences are smaller, as this is close to solar noon (~55°) and therefore far from perpendicular to the wall. Nevertheless, differences are about 2.0 °C. At 15:00 when the sun is in the southwest, differences are up to 1.5-2.0 C in the northeastern corner (figure 6f). Here, it is also evident that parts of the walls are shaded by the trees in this corner, which results in lower $T_{mrt}$ (~ -1.5 °C). When the sun is in west ($\theta \approx 270°$) at 18:00, parts of the southeastern corner are sunlit. This results in ~1.0 °C higher $T_{mrt}$ with the new scheme, compared to the old one. In the last example (21:00), $T_{mrt}$ is lower, ~ -0.6 °C, compared with the old scheme. The sun is close to setting, at an elevation ~5-6°. Thus, the courtyard and its walls are shaded.

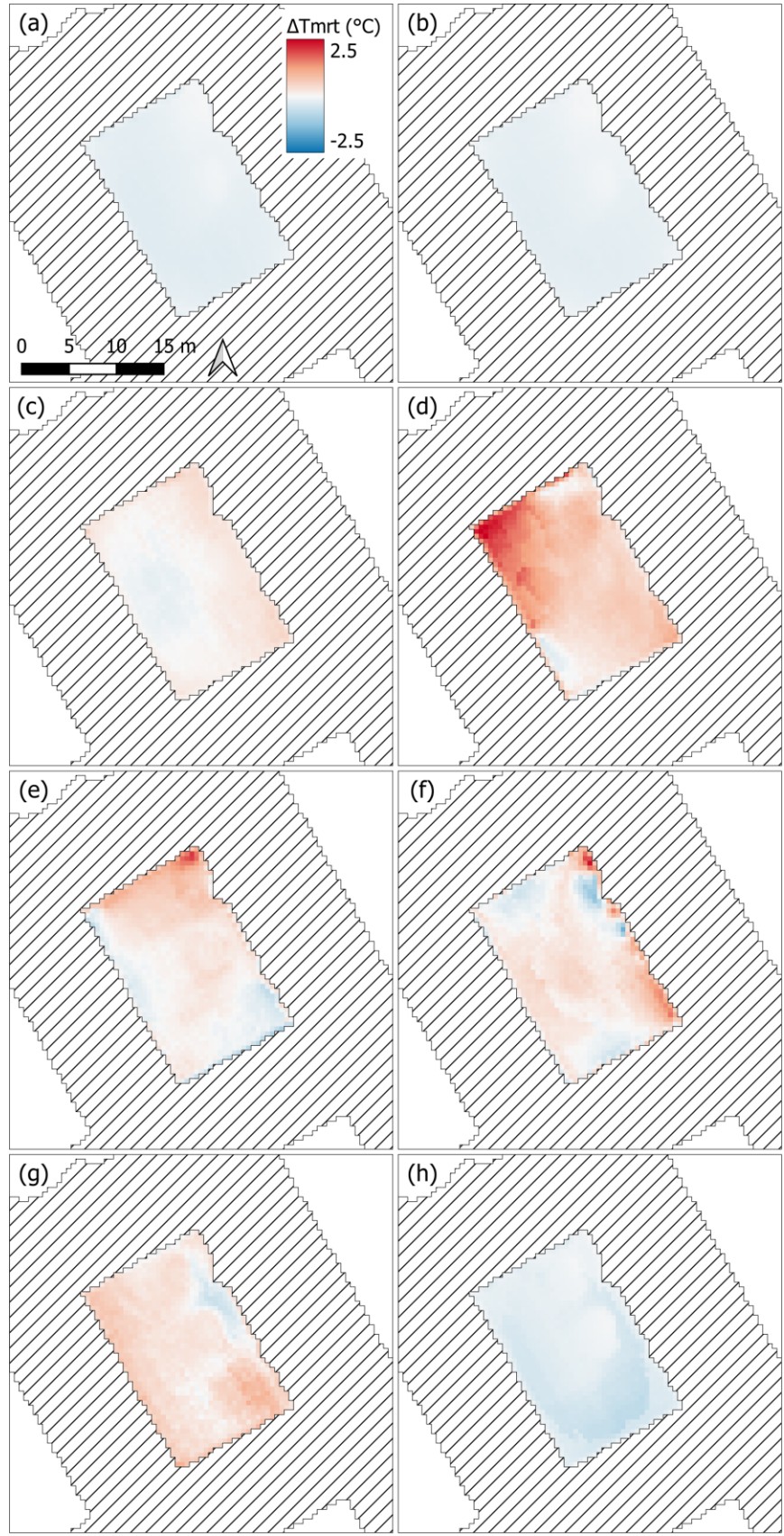

**Figure 6: Difference in mean radiant temperature (T$_{mrt}$) for SOLWEIG simulations with the new and old parameterizations for wall surface temperatures at (a) 2023-06-12 00:00, (b) 03:00, (c) 06:00, (d) 09:00, (e) 12:00, (f) 15:00, (g) 18:00 and (h) 21:00.**

Since the new wall temperature scheme presented here enables calculation of the variation in $T_s$ at different heights, visualization in 3D provides important information on the distribution of $T_s$ throughout the model domain. As an example, Figure 7 presents a 3D visualization of the surface model used in SOLWEIG (Figure 7a) and wall $T_s$ in central Gothenburg on 2023-06-12 17:00 UTC+1 (Figure 7b). The shadow-casting algorithm in SOLWEIG captures shadows from trees, which are visible along the wall in the top right of the figure. In this area, lower wall

temperatures (blue) in shade can be observed, contrasting with the sunlit parts of the wall.

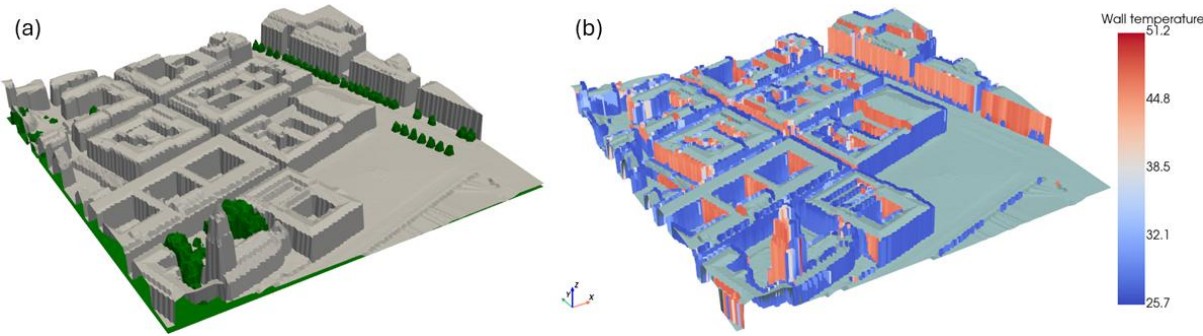

**Figure 7: 3D visualization of the (a) digital surface model (DSM) in grey and canopy digital surface model (CDSM) in green, in the area around Gustaf Adolfs Torg in central Gothenburg, and (b) simulated wall temperatures for the same area. The simulation is for 2023-06-12 17:00:00 UTC+1. Pixel and voxel size is 1 meter. Produced with PyVista (Sullivan**
**et al., 2019).**

To analyze the effect of the input variables of the model on the resulting $T_s$, a sensitivity test was conducted. The test is based on an unobstructed wall facing south forced with meteorological data from 2023-06-13 (see figure 2c). Default values are given in table 2. Default albedo and emissivity were 0.5 and 0.95, respectively.

**Table 2: Default values of thermal conductivity ($\lambda$), density ($\rho$) and specific heat capacity ($C$) for the dummy wall**
**used in the sensitivity test.**

| Material property | Wood | Brick | Concrete |
|---|---|---|---|
| $\lambda$ (W m$^{-1}$ K$^{-1}$) | 0.17 | 0.84 | 1.17 |
| $\rho$ (kg m$^{-3}$) | 700 | 1700 | 2200 |
| $C$ (J kg$^{-1}$ K$^{-1}$) | 1880 | 800 | 840 |

As mentioned in the methods, specific heat capacity and density has no effect on the resulting $T_s$ as any changes in these two variables are cancelled out by corresponding changes in $t$. The remaining input variables, emissivity, albedo, thermal conductivity and wall thickness, influence $T_s$ to different extents. The three first

variables are depicted in figure 8. The influence of wall emissivity is relatively small (figure 8a), and $T_s$ increases with a decrease in emissivity, as expected. There is less radiative cooling and therefore $T_s$ increases. Albedo (figure 8b), on the other hand, influences the amount of absorbed shortwave radiation. As such, $T_s$ increases with a decrease in albedo as more shortwave radiation is absorbed by the wall. A decrease in thermal conductivity (figure 8c), likewise, leads to an increase in $T_s$. Lower thermal conductivity means that less heat is

conducted through the material and instead is used to heat up the surface.

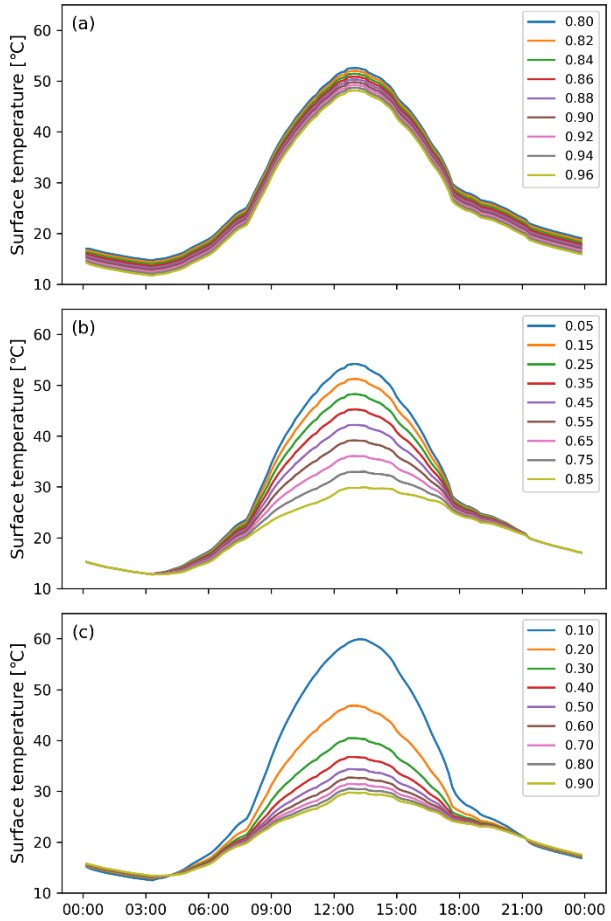

**Figure 8: Simulated wall surface temperatures for an unobstructed wooden wall facing south with different (a) emissivity, (b) albedo and (c) thermal conductivity** (W m$^{-1}$ K$^{-1}$)**.**

The influence of wall thickness on $T_s$ differs from the previous variables as it also depends on the other thermal properties of the material. Three examples are shown in figure 9: wood (figure 9a), brick (figure 9b) and concrete (figure 9c). The wooden wall is very sensitive to changes in wall thickness. As evident in figure 9a, $T_s$ deteriorates already between 0.04 and 0.06 m. For the brick wall, in figure 9b, $T_s$ starts to deteriorate after 0.2 m, indicating that it is less sensitive to changes in thickness. In the last example (figure 9c) the concrete wall starts to deteriorate after 0.5 m.

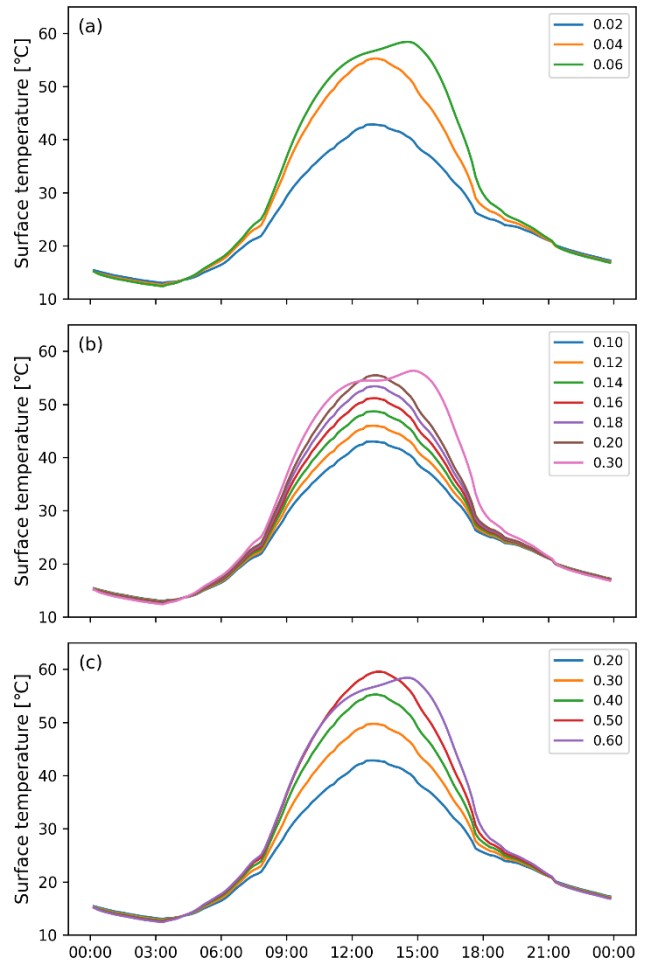

**Figure 9: Simulated wall surface temperatures for an unobstructed wall facing south with different wall thickness (m) for three different materials: (a) wood, (b) brick and (c) concrete.**

The previous figure (figure 9) indicates that $T_s$ starts to deteriorate at different wall thickness depending on material. For example, $T_s$ for wood starts to deteriorate at wall thickness < 0.06 m, brick < 0.3 m and concrete at

0.5-0.6 m. The ratio $R = \frac{1}{e}\sqrt{\frac{t}{\pi}}$ (K m$^2$ W$^{-1}$) against wall thickness is visualized in figure 10. Here it is visible that

these wall thicknesses correspond to $R \approx 0.06$ K m$^2$ W$^{-1}$ for wood and brick. Concrete, that is a substantially more inert material, potentially has a higher threshold for its $R$, but is nevertheless close to 0.06. The ratio $R$ is a good indicator for when the model might start to behave erratically and can be used to warn potential users of too thick walls. Because we are using a characteristic time $\boldsymbol{t}$ with our semi-infinite step heating approach, the

estimated $R$ reduces to a simple function of wall thickness over thermal conductivity, $R = \frac{L}{\pi\sqrt{\pi}\lambda}$, that can be used

to estimate the maximum wall thickness $L_{max} = \pi\sqrt{\pi}\lambda R$. Using a conservative $R = 0.05$ K m$^2$ W$^{-1}$ instead of 0.06 K m$^2$ W$^{-1}$ gives $L_{max} = 0.047, 0.234$ and 0.326 m for wood, brick and concrete, respectively.

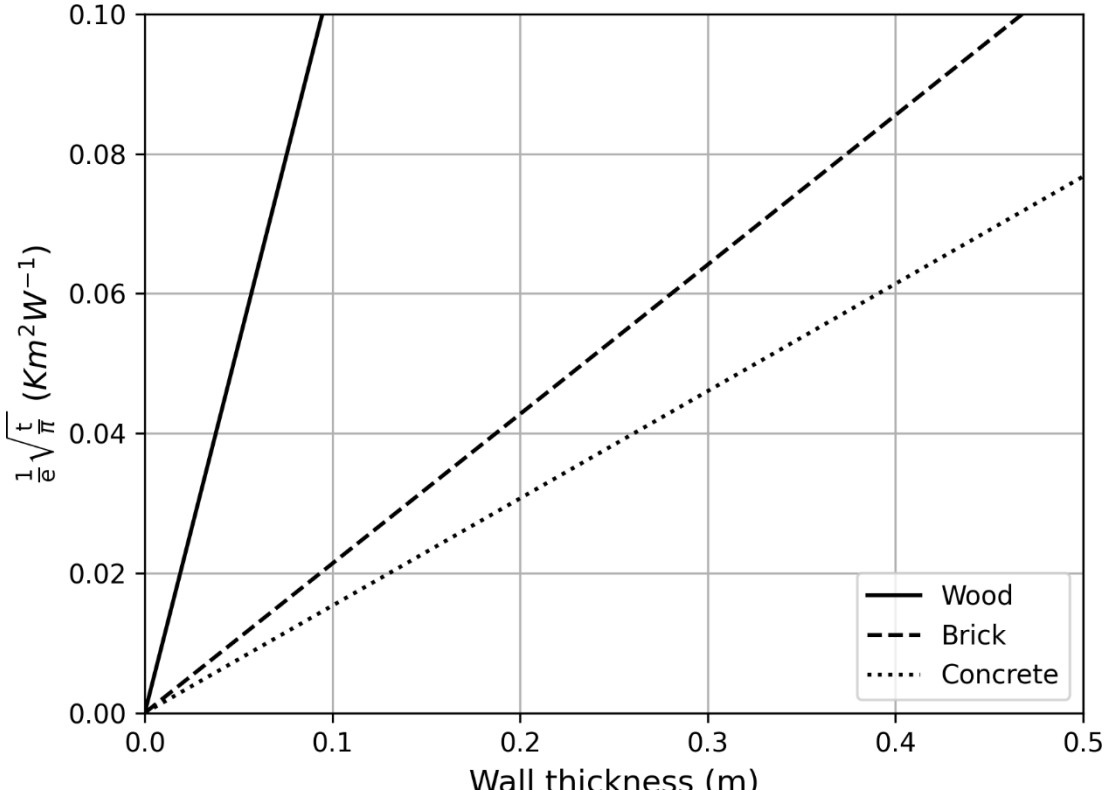

**Figure 10: Relationship between $\frac{1}{e}\sqrt{\frac{t}{\pi}}$ (K m$^2$ W$^{-1}$) and wall thickness (m).**

**4 Discussion**

A new and simple method for simulations of wall $T_s$ has been presented. This new parameterization scheme, based on the works of Boue & Fournier (2009), enables estimations of wall $T_s$ for individual vertical pixels described as voxels. This can be compared to the previous method where wall $T_s$ was estimated for either sunlit or shaded façades (Lindberg et al., 2008).

The five-day examples presented in figure 2 showcase that simulations follow observations. High agreement between simulations and observations is also evident from the statistics in figures 4 and 5, even though correlations decrease with cloudiness, i.e. neither overcast nor clear weather conditions. The lower correlations under cloudy conditions can easily be explained by the fact that the SOLWEIG model does not solve for explicit cloud cover but depends on observations of shortwave radiation, and that observations originate from a weather station located

approximately 2.3 km from the wall of interest for the model evaluation. Nevertheless, $R^2$ values are high (0.93 and 0.94 for wood and plaster brick respectively), and based on 15,394 observation points, covering the entire summer of 2023. RMSE are also low (2.09 and 1.94 °C). These can be compared to RMSE values for simulated $T_s$ of walls on traditional (3.3 °C) and contemporary (7.4 °C) buildings in Dejvice, Prague, Czech Republic, with the PALM-4U model (Resler et al., 2021) and the $T_s$ simulated with the ENVI-met model for a brick wall covered

with plaster ($R^2$=0.98-0.99, RMSE=1.03-1.25) (Simon, 2016). The better performance of the ENVI-met simulations compared to our method could potentially be explained by the fact that i) ENVI-met make use of a 7-node conduction model and ii) their observations are for a controlled building test site, whereas our results are from a real-world example, where we have less knowledge of the thermal properties of the walls. Other examples

of models where our results compare are e.g. TUF3D (Krayenhoff & Voogt, 2008) and SOLENE (Henon et al.,
2012). The temperature range for the wooden wall is larger than that of the plaster brick wall, which could be
explained by the material characteristics. The wooden wall has a lower thermal effusivity compared to the plaster
brick wall, meaning that the wooden wall is more sensitive to changes in radiation. Thus, the $T_s$ of the wooden
wall will fluctuate more, especially during cloudy conditions. The lower thermal effusivity also explains why the
$T_s$ of the wooden wall is higher compared to the plaster brick wall.

The present simulation of wall $T_s$ may look simple since there are only two variables involved ($T_{air}$ and net
radiation of the wall). However, there are several steps to get the net radiation of the wall. The shortwave radiation
depends on the angle between the direct solar radiation and the wall but also reflected radiation from the
surrounding surfaces and shadow patterns. The estimation of the outgoing longwave radiation is problematic since
it requires information on $T_s$ of the wall, i.e. the same variable we aim to simulate. As a proxy we use an iterative
process starting with the $T_s$ from the previous timestep. Already after the first iteration the wall $T_s$ is stabilized
within a few tenths of a degree, which we consider to be close enough to calculate the outgoing longwave
radiation.

The wall surface energy balance is of course not only influenced by the net radiation but also by the sensible heat
flux to/from the air and the heat flow within the wall. Both these processes are forced by the wall $T_s$ as well as
other atmospheric conditions (e.g. wind/turbulence). However, the close connection between the observed and
simulated $T_s$ in our study shows that net radiation dominates heating and cooling of the wall surfaces. A similar
conclusion was made by Resler et al. (2021). They found it striking that the agreement was better between
observed and modelled wall $T_s$ during summer episodes with strong radiative forcing. However, the observed wall
$T_s$ of plastered brick walls both in our study and in the study by Resler et al. (2021) were higher than the simulated
after the daytime maximum, which indicates an influence of heat stored in the wall earlier during the day. Higher
observed than simulated minimum wall $T_s$, as also noted by Resler et al. (2021), point to an addition of stored heat
to the wall surface. On the other hand, the thinner wooden wall with less ability to store heat was cooler than the
simulation during the night.

The new parameterization scheme has enabled better estimations of wall $T_s$ and introduced estimations of shading
on walls from surrounding buildings and trees (e.g. Figure 7b). This results in differences in $T_{mrt}$ between this new
version and previous versions of SOLWEIG. Some of these differences are presented in figure 6 and show that
the location of the sun has an effect on $T_{mrt}$. This is explained by the estimation of wall $T_s$ for individual voxels,
where each voxel has its own relation to the location of the sun because of its material characteristics (thermal
effusivity, albedo, emissivity) but not the least aspect. The aspect of the wall, i.e. its orientation, influences how
much of the direct shortwave radiation that it is exposed to. Previous studies have concluded that the old
parameterization in SOLWEIG led to deviations in simulated wall $T_s$ that affected $T_{mrt}$ (Gal & Kantor, 2020;
Wallenberg et al., 2023).

Gal and Kantor (2020), using SOLWEIG, saw deviations in simulated $T_{mrt}$ in proximity to wall surfaces and
suggested a separate surface temperature parameterization scheme. Some of the overestimations were potentially
reduced by around 3 °C with the introduction of patches for short- and longwave radiation described in Wallenberg
et al. (2020; 2023a). The new parameterization scheme for wall $T_s$ should improve these deviations further. For

example, underestimations were seen in vicinity of sunlit areas (Gal & Kantor, 2020). Here, we show (e.g. Figure 6d) that the new wall surface scheme calculates higher $T_{mrt}$ close to sunlit walls (up to 2.5 °C for our case). A small decrease is also noticeable under the trees (Figure 6f-g). These differences come from the directionality of the walls and accounting for whether they are sunlit or shaded, which was previously not possible.

Kim & Ham (2024) found deviations in simulated $T_s$ of asphalt, plywood and soil, and suggested improvements in thermophysical attributes of construction materials in calculations in SOLWEIG. Although their study referred to horizontal surfaces (e.g. ground and roofs) the suggestions are in line with what has been presented in this paper. The old parameterization scheme for wall $T_s$ is based on the parameterization scheme for ground surface temperatures utilized in SOLWEIG. This scheme was initially developed by Bogren et al. (2000) and refined by Lindberg et al. (2016) and is based on a sinusoidal curve that peaks at a set time. For ground surfaces, this peak in time might be accurate as it occurs not long after solar noon. For walls, however, one peak time for the entire model domain will lead to differences in $T_s$, depending on the aspect of the wall. Peak time of a wall facing east will for example occur in the morning, whereas it will occur in the afternoon or evening for a wall facing west. The $T_s$ scheme presented here, thus, is an improvement in SOLWEIG, but for wall surfaces. The method presented could, however, be useful for ground surfaces as well.

Even though the model performs well, there are some shortcomings. For example, the spatial input data is pixel based. This leads to deviations when estimating wall aspects which could be seen in figure 7b (stripe effect). Although the accuracy of aspect calculations is high in mid-sections of the walls, deviations occur close to corners. Other potential sources of error could be the material characteristics used here, i.e. thermal conductivity, wall thickness, emissivity and albedo. In addition to this, there is a possibility that there are deviations based on longwave radiation reflected from the background, i.e. the opposite of our sensors (Apogee Instruments, 2024b), something that is difficult to estimate in urban areas with many different surfaces that have different emissivity. In relation to the sensors, there is also a possibility that there are micro-shadows on e.g. the wooden wall from the microscale spacing between the boards. Other shortcomings that are likely to influence our results are that we are modeling environmental radiative forcings at the surface of the wall, i.e. input short- and longwave radiation. As they are simulated, they are not perfect, with impacts on the resulting $T_s$. This also goes for the properties of the wall surfaces, e.g. albedo, emissivity, thermal conductivity and wall thickness. Similar conclusions are described in Johnson et al. (1991), where their simulated rural and urban surface temperatures performed better when the specification of input thermal properties and environmental forcings were most complete and accurate. Similar conclusions can be given about the basis of our step heating approach in Boue & Fournier (2009), where they have complete control over the forcing (heat pulse), thermal properties and not the least environmental conditions of their lab setting.

It is evident from the sensitivity analysis that there are constraints in the model. For example, there are limits to how thick a wall can be, depending on its thermal conductivity (fig. 9). The presented solution, using $R$ to estimate $L_{max}$ to get an indication on when the model might become unstable can hopefully guide users (e.g. fig. 10). In addition to this, the sensitivity analysis shows that albedo and thermal conductivity have largest effects on $T_s$ and emissivity least (fig. 8). These results are expected. An increase in albedo should lead to a decrease in $T_s$ as less shortwave radiation is absorbed. An increase in thermal conductivity leads to more energy propagating through the material and less energy at its surface, i.e. lower $T_s$. A material with higher emissivity has higher radiant

cooling and therefore a lower $T_s$. Thermal conductivity and emissivity are material properties that should affect the thermal comfort of a human, i.e. if they are changed, so is the exposure to a human. A change in albedo, on the other hand, leads to less absorption of heat by a surface (if it is increased), i.e. decrease in longwave radiation, and increases the reflectivity of shortwave radiation. Thus, a human would be exposed to less longwave radiation but more shortwave radiation (Erell et al., 2014).

The new scheme has only introduced four new input parameters to the SOLWEIG model: thermal conductivity, density, specific heat capacity and thickness of the wall. These parameters are easily accessible in literature (e.g. CIBSE, 2015). Three materials are available in the new publicly available SOLWEIG model: brick, concrete and wood. Other materials can simply be added to the parameter file in SOLWEIG. One drawback with the new scheme is the computational complexity. Running the model for one day at 10-minute resolution (144 time steps) took approximately 5 minutes with the newly implemented wall scheme parameterization, compared to around 1 minute without it, on an Intel i7-7700 CPU @ 3.6 GHz with 16 GB RAM. However, this is still faster than most other models designed for similar purposes.

Future efforts will include improvements of the ground surface calculations. Initial results indicate that the force-restore method is a well-suited framework for this application. Moreover, the parameterization scheme should be evaluated for other wall surfaces and urban environments, as well as a deeper evaluation of the behavior of walls and their influence on the microclimate and thermal comfort of humans.

**5 Conclusions**

We have presented a new parameterization scheme for calculations of wall $T_s$. The scheme is simple and fast, with only four new input parameters: thermal conductivity, density, specific heat capacity and wall thickness, together with a heat flux density, in this case the heat balance of the wall surface. The performance of the scheme is high for the two assessed wall surfaces, wood ($R^2 = 0.93$, RMSE = 2.09 °C) and plaster brick ($R^2 = 0.94$, RMSE = 1.94 °C), making it comparable to heat conductivity models utilized in other models for microclimate.

The implementation of the scheme in the SOLWEIG model has enabled more realistic estimations of wall $T_s$ by incorporating wall aspects and material characteristics. In the examples presented here this has led to differences in $T_{mrt}$ of up to 2.5 °C compared to the previous version of SOLWEIG.

With this new scheme SOLWEIG can now be used to understand how different building materials influence outdoor thermal comfort. The high accuracy indicated by our results combined with the computational speed suggests that this approach can be used in other areas where $T_s$ of walls are important, for example building energy and urban energy balance models.

The new version of SOLWEIG presented here and its associated datasets are available in Wallenberg et al. (2025a; b).

**Code availability**

The mode code is available from https://doi.org/10.5281/zenodo.15309383 (Wallenberg et al., 2025a). A dataset consisting of required forcing data for SOLWEIG (DEM, DSM, CDSM, ground cover, meteorological data) and observed and simulated wall surface temperatures used in the study are available from https://doi.org/10.5281/zenodo.15309444.

**Author contributions**

NW developed the model code (Software) and performed the simulations (Formal analysis). NW, FL, and JL prepared and conducted the field campaign (Investigation). NW prepared the manuscript (Writing – original draft) with contributions from all co-authors (Writing – review & editing). EM and DR formulated the initial model design (Conceptualization), and NW, BH, FL, and JL finalized the model (Methodology).

**Competing interests**

The authors declare that they have no conflict of interest.

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
