# Peer review of "A simple step heating approach for wall surface temperature estimation in the SOLar and LongWave Environmental Irradiance Geometry (SOLWEIG) model"

_EGUsphere, 2025_

## Referee Comment (RC2)

**Manuscript egusphere-2025-2093**

"A simple step heating approach for wall surface temperature estimation in the SOlar and LongWave Environmental Irradiance Geometry (SOLWEIG) model"

by

Nils Wallenberg, Björn Holmer, Fredrik Lindberg, Jessika Lönn, Erik Maesel, David Rayner

**Review**

The interesting manuscript consists of an informative abstract, a clearly structured, easily readable text, a table, seven figures (with up to eight subfigures), and a list of 51 references. All figures (three in black and white, four in color) and the table are essential for a deeper understanding of the text and therefore cannot be deleted.

The manuscript deals with an issue that is currently of great importance in the context of developments of radiation models, which are necessary for different applications in urban meteorology, e.g., the implementation of strategies focusing on the reduction of human heat stress in outdoor urban spaces. The SOLWEIG model, to which this study refers, has already proven to be very effective in previous studies.

Both in terms of content and appearance, the manuscript makes an excellent impression on the reviewer. Only two quotes remain to be discussed:

- Page 2, line 39: PET was first introduced in Mayer and Höppe (1987). Therefore, this
  citation should be added to Höppe (1999) or replace Höppe (1999). Höppe et al. (1999)
  is incorrect.
  - Mayer, H. and Höppe, P.: Thermal comfort of man in different urban environments. Theor. Appl. Climatol. 38(1), 43-49, https://doi.org/10.1007/BF00866252, 1987.
- Page 2, line 39: The quote about UTCI (Blazejczyk et al., 2010) is correct. However, based on the paper by Lee et al. (2025), it should be noted here that, for physical reasons, the UTCI cannot be applied in urban open spaces, i.e., within the urban canopy layer.
  - Lee, H., Park, S., and Mayer, H.: Approach for the vertical wind speed profile implemented in the UTCI basics blocks UTCI applications at the urban pedestrian level. Int. J. Biometeorol. 69, 567-580, https://doi.org/10.1007/s00484-024-02835-x, 2025.

**Review according to template**

**Scientific significance:**

Does the manuscript represent a substantial contribution to modelling science within the scope of Geoscientific Model Development (substantial new concepts, ideas, or methods)?

**Evaluation: excellent (1)**

**Scientific quality:**

Are the scientific approach and applied methods valid? Are the results discussed in an appropriate and balanced way (consideration of related work, including appropriate references)? Do the models, technical advances, and/or experiments described have the potential to perform calculations leading to significant scientific results?

**Evaluation: excellent (1)**

**Scientific reproducibility:**

To what extent is the modelling science reproducible? Is the description sufficiently complete and precise to allow reproduction of the science by fellow scientists (traceability of results)?

**Evaluation: excellent (1)**

**Presentation quality:**

Are the methods, results, and conclusions presented in a clear, concise, and well-structured way (number and quality of figures/tables, appropriate use of English language)?

**Evaluation: excellent (1)**

In the full review and interactive discussion, the referees and other interested members of the scientific community are asked to take into account all of the following aspects:

- Does the paper address relevant scientific modelling questions within the scope of GMD? Does the paper present a model, advances in modelling science, or a modelling protocol that is suitable for addressing relevant scientific questions within the scope of EGU? yes, sure
- 2. Does the paper present novel concepts, ideas, tools, or data? yes, sure
- Does the paper represent a sufficiently substantial advance in modelling science?
   yes, sure
- 4. Are the methods and assumptions valid and clearly outlined? yes, sure
- 5. Are the results sufficient to support the interpretations and conclusions?
  yes, sure
- 6. Is the description sufficiently complete and precise to allow their reproduction by fellow scientists (traceability of results)? yes, sure
  - In the case of model description papers, it should in theory be possible for an independent scientist to construct a model that, while not necessarily numerically identical, will produce scientifically equivalent results. Model development papers should be similarly reproducible. For MIP and benchmarking papers, it should be possible for the protocol to be precisely reproduced for an independent model. Descriptions of numerical advances should be precisely reproducible.

    everything is ensured
- 7. Do the authors give proper credit to related work and clearly indicate their own new/original contribution? **yes, sure**
- 8. Does the title clearly reflect the contents of the paper? The model name and number should be included in papers that deal with only one model. **yes, sure**
- 9. Does the abstract provide a concise and complete summary? yes, sure
- 10. Is the overall presentation well structured and clear? yes, sure
- 11. Is the language fluent and precise? yes, sure
- 12. Are mathematical formulae, symbols, abbreviations, and units correctly defined and used? **yes, sure**
- 13. Should any parts of the paper (text, formulae, figures, tables) be clarified, reduced, combined, or eliminated? **no**
- 14. Are the number and quality of references appropriate? see comments on page 1

15. Is the amount and quality of supplementary material appropriate? For model description papers, authors are strongly encouraged to submit supplementary material containing the model code and a user manual. For development, technical, and benchmarking papers, the submission of code to perform calculations described in the text is strongly encouraged. The notes in the manuscript on code availability are entirely sufficient.